# Contrastive-Equivariant Self-Supervised Learning Improves Alignment with Primate Visual Area IT

**Thomas Yerxa** [1] [*]    **Jenelle Feather** [1,2]    **Eero P. Simoncelli** [1,2]    **SueYeon Chung**[1,2]

[1]Center for Neural Science, New York University
[2]Center for Computational Neuroscience, Flatiron Institute, Simons Foundation

## Abstract

Models trained with self-supervised learning objectives have recently matched or surpassed models trained with traditional supervised object recognition in their ability to predict neural responses of object-selective neurons in the primate visual system. A self-supervised learning objective is arguably a more biologically plausible organizing principle, as the optimization does not require a large number of labeled examples. However, typical self-supervised objectives may result in network representations that are overly invariant to changes in the input. Here, we show that a representation with structured variability to input transformations is better aligned with known features of visual perception and neural computation. We introduce a novel framework for converting standard invariant SSL losses into "contrastive-equivariant" versions that encourage preservation of input transformations without supervised access to the transformation parameters. We demonstrate that our proposed method systematically increases the ability of models to predict responses in macaque inferior temporal cortex. Our results demonstrate the promise of incorporating known features of neural computation into task-optimization for building better models of visual cortex.

## 1 Introduction

In the past decade, task-optimized deep neural networks (DNNs) have been used to predict responses of object-selective neurons in primates to natural image stimuli [Yamins et al., 2014, Schrimpf et al., 2020, Willeke et al., 2023]. Such networks have a pronounced advantage over more traditional models for explaining responses in deeper areas with more abstract representations, such as inferior temporal cortex (IT). This observation naturally leads to the hypothesis that task optimization can provide a normative account for IT neuron tuning properties: late-stage visual representations are shaped by the need to perform ecologically relevant tasks.

However, the task that initially led to these advances was that of supervised object classification, a specific task that relies on an implausibly large number of labeled examples [Lindsay, 2021]. More recently, computer vision has undergone a "self-supervised learning" (SSL) revolution. A variety of methods have been proposed to learn representations that match or surpass supervised training on multiple tasks by deriving sources of supervision from the data itself rather than relying on human annotations. For example, many popular SSL strategies aim to unify representations of different transformations of the same image (commonly referred to as "views"), while enforcing diversity among representations of distinct images. Additionally, self-supervised representations can predict primate neural responses with fidelity comparable to supervised representations [Zhuang et al., 2021, Konkle and Alvarez, 2022, Parthasarathy et al., 2024].

---

[*]Corresponding Author: `tey214@nyu.edu`

38th Conference on Neural Information Processing Systems (NeurIPS 2024).

Both of these training objectives are forms of invariance learning: responses of an ideal object classification model should be invariant across different objects from the same class, and self-supervised learning strives to achieve invariance to the transformations used to generate different views. However biological visual representations are not fully invariant across views [DiCarlo and Cox, 2007, Kuoch et al., 2024]. Indeed it has been demonstrated that training according to either of these two objectives leads to representations that are invariant to stimulus perturbations that are salient to human observers [Feather et al., 2023]. Additionally, even in Area IT, which is thought to subserve invariant object recognition, neural populations encode a significant amount of "category orthogonal" information (e.g., object pose or viewing conditions that are unrelated to semantic category) [Hong et al., 2016]. Furthermore, such selectivity for object-orthogonal attributes is meaningfully organized within Area IT [Hong et al., 2016] (i.e. object orthogonal attributes are linearly decodable from population responses). Whether such structured variability emerges in invariance-trained networks is likely determined by the uncontrolled inductive biases of the network architecture [Alleman et al., 2024].

Here, we develop an equivariant learning framework that encourages such structured variability in network representations. Our contributions are:

- We propose a novel framework that converts standard invariance-based self-supervised learning methods into "contrastive-equivariant" versions that produce structured, transformation-related variability. Unlike previous approaches, our method does not require supervised access to transformation parameters or costly modifications to the training procedure.

- We examine the tradeoff between invariance and structured variability through a series of representational analyses. We find that, relative to networks trained for invariance alone, our contrastive-equivariant network learns structured transformation variability that is shared across images and factorized with respect to variability related to changes in image content.

- We explore the impact of including an equivariant loss for predicting neural activity in IT, showing for the first time that explicitly encouraging structured variability via optimization leads to an improved ability to predict cortical responses to natural images.

## 2 Method

### 2.1 Transformation-Invariant Self-Supervised Learning (iSSL)

The influential work of [Chen et al., 2020] showed that applying two random transformations (often called "augmentations") to a batch of images, then training a network to identify which pairs of transformed images originated from the same sample with a cross-entropy style loss (the InfoNCE loss, first formulated in [Gutmann and Hyvärinen, 2010]) yields representations that are competitive with supervised training for object classification. Many subsequent studies have developed alternative objective functions that produce similar results: Barlow Twins[Zbontar et al., 2021], VICReg [Bardes et al., 2021], and W-MSE [Ermolov et al., 2021] enforce augmentation invariance along with a constraint that the global covariance matrix is the identity; SimSiam [Chen and He, 2021] and BYOL [Grill et al., 2020] employ architectural constraints that regularize towards uniform representations and simply optimize for transformation invariance. Other studies have formalized the problem in terms of maximizing information [Ozsoy et al., 2022] or capacity [Yerxa et al., 2024], subject to an invariance constraint, which has enabled connections to normative theories of coding efficiency and manifold capacity [Barlow et al., 1961, Chung et al., 2018].

To formalize the definition of iSSL, we denote a dataset of images (e.g., ImageNet) by $X \in \mathbb{R}^{N \times D}$, where $N$ is the number of images and $D$ is their dimensionality (number of pixels). Let $\tau(\cdot; \rho) : \mathbb{R}^D \to \mathbb{R}^D$ be a function parameterized by $\rho$ that maps images to images (for example, for $\tau$ a random crop operation, $\rho$ specifies the region to be cropped). The goal of iSSL algorithms is to learn the parameters $W$ of some function $f(\cdot; W) : \mathbb{R}^D \to \mathbb{R}^d$ such that the variability over $\rho$ is minimal while preserving variability over $X$ (to avoid trivial solutions such as $f(\cdot; W) = 0$ for all inputs). Many methods achieve this by observing pairs of randomly augmented views of a batch of images: $X^A = \tau(X; \rho_1)$, $X^B = \tau(X; \rho_2)$, with $\rho_1, \rho_2 \sim p(\rho)$ where $p(\rho)$ is a pre-chosen probability distribution over augmentation parameters. Generally iSSL frameworks employ an objective function that operates on the outputs of $f$, $Z^A = f(X^A; W)$, $Z^B = f(X^B; W)$. One popular framework is

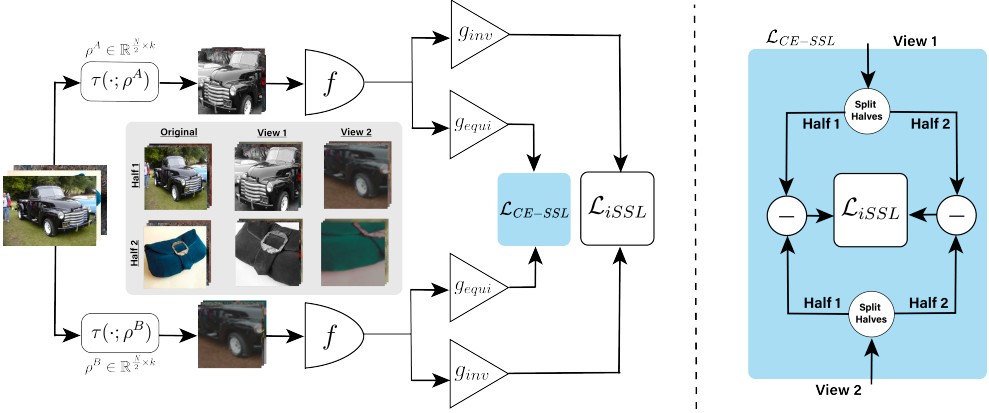

Figure 1: Diagram of the proposed training method. At the beginning of each training epoch the dataset is randomly split into two non-overlapping halves. Left Gray Panel: corresponding images in each subset are augmented using the same set of two random transformations (so the total number of random transformations is halved relative to a standard iSSL training scheme). Every view is passed through a representation network $f$ (ResNet-50 in this work) and the outputs are projected into two embedding spaces by different projector networks, $g_{inv}$ and $g_{equi}$. In the invariant embedding space a standard iSSL loss is applied, while in the equivariant embedding space the same iSSL loss is applied to the difference vectors between transformation-positive pairs (visualized on the right).

"Barlow Twins" [Zbontar et al., 2021], which uses the objective: $\mathcal{L}_{BT} = \Sigma_i (1 - \mathcal{C}_{ii})^2 + \lambda \Sigma_{i, i \neq j} (\mathcal{C}_{ij})^2$ where $\mathcal{C}$ is the cross-correlation matrix between $Z^A$ and $Z^B$. The first term encourages the outputs in response to the same image subject to different augmentations to be correlated, while the second encourages the outputs in response to distinct images to be uncorrelated.

Because complete invariance to the transformations employed in iSSL is harmful for downstream tasks, most frameworks employ a learnable "projector network" that maps the outputs of the representation network to an embedding space before applying the loss. The nearly ubiquitous use of this "guillotine regularization" [Bordes et al., 2022], means that most iSSL methods aim to learn a function *from which an augmentation invariant subspace can be extracted*. While this approach does permit some transformation-related variability in the representation, there is no explicit control or encouragement of that variability, and no incentive for that variability to be usefully structured.

## 2.2 Contrastive-Equivariant Self-Supervised Learning (CE-SSL)

To induce structured variability in learned representations, we require that an equivariant subspace can be extracted from $f$ alongside the invariant subspace described above. A function is equivariant to a set of input transformations if there exists a corresponding set of output transformations that induce the same changes. In the self-supervised learning setting this property can be expressed as:

$$\forall \tau_\rho \in P, \ \forall x \in X, \ \exists T_\rho : f(\tau_\rho(x)) = T_\rho(f(x)), \tag{1}$$

where $\tau_\rho = \tau(\cdot; \rho)$ and $P$ is the set of possible values of transformation parameters. Note that invariance is a special case of equivariance, in which $T_\rho$ is an identity transformation for all $\rho$. To avoid this degenerate solution, we will require both that similarly transformed inputs be related by the same transformation in the output space, and that differently transformed inputs are related to each other by different transformations.

Our training methodlogy, summarized in Fig. 1, follows the principle first proposed by [Gupta et al., 2023]: "Equivariance should be learned from pairs of data, as in invariant contrastive learning." First we split our dataset of images into two random non-overlapping equal-sized partitions $X_1, X_2 \in \mathbb{R}^{\frac{N}{2} \times D}$. Next we apply a randomly selected augmentation to both $X_1$ and $X_2$, so that corresponding rows of $X_1^{A/B}$ and $X_2^{A/B}$ contain distinct images that have been subjected to the same augmentations.

Note that this reduces the total number of random samples of $\rho$ by a factor of two relative to standard iSSL methods. Finally the resulting representation vectors are each fed through two distinct projector networks, $z_i^{A/B} = g_{inv}(r_i^{A/B})$ and $\tilde{z}_i^{A/B} = g_{equi}(r_i^{A/B})$. These two embeddings are optimized to be invariant to transformations and discriminative across base images, or invariant to base images and discriminative across transformations, respectively. The overall objective (loss) functions is:

$$\mathcal{L}_{\text{overall}} = (1 - \lambda)\mathcal{L}_{iSSL} + \lambda\mathcal{L}_{CE-SSL}$$
$$\text{where} \quad \mathcal{L}_{iSSL} = \mathcal{L}([z_1^A, z_2^A], [z_1^B, z_2^B]), \tag{2}$$
$$\text{and} \quad \mathcal{L}_{CE-SSL} = \mathcal{L}(z_1^A - z_1^B, z_2^A - z_2^B),$$

where both terms are written in terms of $\mathcal{L}$, a self-supervised learning loss function that encourages invariance and uniformity [Wang and Isola, 2020] (e.g., $L_{BT}$) and $\lambda$ is a hyperparameter that determines the relative importance of extracting an invariant or equivariant subspace from the shared representation. In the notation of Eq. (1), by designing $\mathcal{L}_{CE-SSL}$ to encourage similar transformations to induce similar displacements in the output space, we are implicitly specifying that our output transformations are of the form $T_\rho(z) = z + z_\rho$. Thus we leverage the principles underpinning contrastive invariance learning to encourage representations that contain useful transformation-related information; this choice differentiates this formulation from previous equivariant self-supervised learning approaches.

## 3 Results

### 3.1 Implementation Details

**Architecture and invariance ojective.** For all experiments we use a ResNet-50 architecture [He et al., 2016] as the backbone representation network $f$. Our training scheme is compatible with any choice of iSSL framework, as specified by the choice of $\mathcal{L}_{iSSL}$. We experimented with three different base methods chosen to span the range from "instance contrastive" to "dimension contrastive" [Garrido et al., 2023a]: SimCLR [Chen et al., 2020], MMCR [Yerxa et al., 2024], and Barlow Twins [Zbontar et al., 2021]. In each case, we define $g_{inv}$ using the projector network architecture proposed in the original work. To retain the synergy between the normalization scheme, loss function, and projector architecture achieved by each framework we use the same architecture for both $g_{inv}$ and $g_{equi}$.

**Pretraining dataset and augmentations.** We train using the ImageNet-1k dataset and the standard set of augmentations first introduced in [Grill et al., 2020], which includes random resized cropping, color jittering, Gaussian blurring, solarization, and horizontal flips. See Appendix A.2 for exact training details.

**Invariance-equivariance tradeoff.** For each of the three choices of $\mathcal{L}_{iSSL}$ we trained networks with hyperparameter values $\lambda \in \{0.0, 0.001, 0.1, 0.2, 0.3, 0.4, 0.5\}$, yielding a total of 21 learned representations (note: $\lambda = 0$ corresponds to standard iSSL). We found that classification performance becomes severely degraded for values of $\lambda$ larger than $0.5$ (see Appendix A.4).

### 3.2 Representational Analyses

**Bures metric comparisons.** We conducted a series of experiments to determine the extent to which various sources of variability in our dataset were meaningfully organized. The experiments utilized the Bures metric, which is the Wasserstein ("Earth Mover's") distance between mean-centered Gaussian distributions with covariance matrices $C_1$ and $C_2$:

$$D_B(C_1, C_2) = \text{trace}\left(C_1 + C_2 - 2\left(C_2^{1/2}C_1C_2^{1/2}\right)^{1/2}\right). \tag{3}$$

When $C_1$ and $C_2$ are normalized to have a trace of 1, the maximal distance of $2.0$ occurs when the variabilities lie in orthogonal subspaces (or are completely "factorized" from each other) and the minimum distance of $0.0$ occurs when the covariances are equal. More generally, a large Bures distance indicates two sources of variability are factorized from each other and a low distance

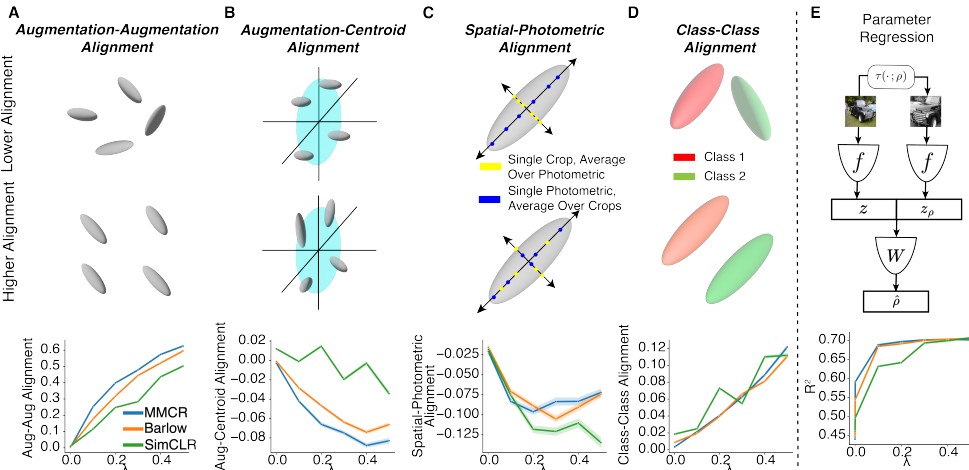

Figure 2: Effects of equivariance on representational geometry. **A**: Alignment between augmentation manifolds (gray ellipsoids). **B**: Alignment between augmentation manifolds (gray ellipsoids) and the centroid manifold (blue disk). **C**: Alignment between spatial and photometric manifolds. Gray ellipsoids represent single augmentation manifolds, and blue/yellow points indicate the mean over the outputs from many transformations of a single view obtained via a photometric/spatial transformation, respectively. Expected distance is larger when the two sources of variability are factorized. **D**: Same as A., but for class manifolds. **E**: A schematic of the parameter regression experiment. In each panel, the bottom row depicts the results of each analysis described in the text of Sections 3.2 and 3.2. Shaded regions indicate 95% confidence intervals (estimated over the same comparisons the expected distance is estimated over for A-D and over 5 independent runs of the regression experiments for E). A summary of the sources of variability used to compute $C_1$ and $C_2$, and the ensemble used to estimate the expected alignment is measured can be found in Table A.5

indicates shared structure. We first estimate the trace-normalized covariance of the outputs of some network $f$ over two sources of variability and compute the Bures metric between the two.

Because we are mainly interested in the impact of the equivariance loss relative to the invariant baseline, each analysis below is carefully controlled to expose any structural differences. In particular, we estimate $C_1$ and $C_2$ over identical inputs for an invariant network and an equivariant network trained using the same base objective but a non-zero value of $\lambda$. We then can directly compare the measured Bures distance for the invariant network and each equivariant network ($\lambda \neq 0$): $\Delta D_B = D_B(C_1^{\lambda=0}, C_2^{\lambda=0}) - D_B(C_1^{\lambda\neq0}, C_2^{\lambda\neq0})$; this measure quantifies the amount of alignment between two sources of variability in an equivariant network relative to the invariant network baseline. In each of the panels in the bottom row of Fig. 2 we show how $\mathbb{E}[\Delta D_B]$ (denoted simply "Alignment") evolves as a function of $\lambda$ when $C_1$ and $C_2$ are estimated over different sources of variability (y-axis labels indicate the ensembles over which the variabilities were estimated). We show the joint distribution of $(D_B(C_1^{\lambda=0}, C_2^{\lambda=0}), D_B(C_1^{\lambda\neq0}, C_2^{\lambda\neq0}))$ and summarize the sources of variability in each experiment described below in Appendix A.5.

**Augmentation-Augmentation alignment.** First we determine the extent to which augmentation variability is shared across base images in each network (Fig. 2A). For these experiments, both $C_1$ and $C_2$ are estimated over many random transformations of single images in the validation set (we will refer to the responses to such a group as an "augmentation manifold"). The expectation is then over randomly sampled pairs of augmentation manifolds. The positive expected difference of distance indicates the equivariant networks consistently produce lower distance between augmentation manifolds, indicating more shared augmentation variability across base images. This structure is closely related to what is encouraged by the equivariance loss term and the orderly increase as a function of $\lambda$ suggests that we are optimizing effectively.

**Augmentation-Centroid factorization.** We next investigate the extent to which variability over augmentations is factorized from variability over base images (Fig. 2B). We use the "centroid

manifold," [Yerxa et al., 2024] to characterize the variability over base images, by measuring the covariance over base images of the means over augmentations. That is, for these experiments (for each network respectively) $C_1$ is the covariance of the centroids of all augmentation manifolds and $C_2$ is the covariance of a randomly selected augmentation manifold. We observe that equivariant networks generally exhibit a larger distance between centroid and augmentation manifolds indicating increased factorization (or lower alignment) of image-content variability and image-augmentation variability. This structure was not explicitly encouraged by the objective and can be considered an emergent property of the equivariant learning procedure.

**Spatial-Photometric factorization.**    Next we ask whether our equivariant training procedure induced increased factorization of variability to different types of input transformations (Fig. 2C). The standard augmentation procedure involves first taking a random crop (spatial variability) of a given image and then applying a series of pixel-level transformations (color-jittering, gaussian blurring, etc.) (photometric variability). To assess the impact of these two distinct classes of image transformations we first chose 20 random crops a given image, then applied the same set of 20 random photometric transformations to each individual crop, yielding 400 different views of each base image. $C_1$ and $C_2$ are then estimated over network responses that are averaged over different crops or different photometric transformations respectively, and the expectation is taken over different (single) base images. We observe the equivariant networks consistently exhibit increased factorization (i.e. larger Bures distances relative to the invariant trained network). This again is an emergent property of the equivariant learning procedure, and is particularly interesting in light of recent work that discovered that this form of transformation-factorization is more correlated with neural predictivity than transformation invariance [Lindsey and Issa, 2024].

**Class-Class factorization.**    Finally we asked whether within-class variability was more or less shared between distinct classes in equivariant networks by estimating $C_1$ and $C_2$ over responses to all images in distinct classes in the validation set (the expectation is then taken over different random pairs of classes) (Fig. 2D). Increased sharing of variability between class manifolds has been demonstrated to increase manifold capacity, and can make representations better suited for multi-task evaluations [Wakhloo et al., 2023, 2024]. We observe higher alingment (lower expected pairwise Bures distances) in the equivariant networks indicating that the "class manifolds" relative to the invariant networks.

**Linear embedding of augmentation-related information.**    While the above experiments demonstrate that equivariant training induces increased alignment of transformation-related variability between images, this does not necessarily imply that this variability is coherently organized. To assess this more directly, we measure the extent to which augmentation parameters can be linearly decoded from the networks' representations. Specificlly, we regress the concatenated outputs of a clean and transformed image onto the parameters of the applied augmentation. We report the resulting coefficient of determination ($R^2$) on a heldout set of validation images (Fig. 2). The equivariant training is seen to increase the amount of linearly accessible augmentation information relative to invariant training (the leftmost points plotted in Fig 2E). We further analyzed a set of equivariant models trained with weaker augmentation parameters (see A.6 for details). In these networks, we again observe that equivariant training increased the amount of linearly accessible augmentation information compared to invariant training. This holds not only for augmentation parameters within the training range (left panel 7) but also for parameter values beyond the training range (right panel 7). Thus, the equivariance properties of the models generalize beyond the training distribution. Future work could examine generalization to unseen types of augmentations.

### 3.3    Neural Predictivity

We utilized the BrainScore evaluation pipeline Schrimpf et al. [2018] to measure the extent to which each learned representation can linearly predict neural responses measured in macaque area IT, for four different experimental datasets. At the time of testing, our highest performing model (Barlow Twins objective, $\lambda = 0.2$) had the 10th highest average predictivity for area IT out of approximately 250 publicly available models on the Brain-Score leaderboard. Across a reasonably large range of values of $\lambda$, the equivariant models improved the neural predictivity relative to the invariant baseline ($\lambda = 0$) for all four datasets (Fig. 3). Many previous publications have noted that changes in training objective function have a small effect on neural predictivity, relative to other factors

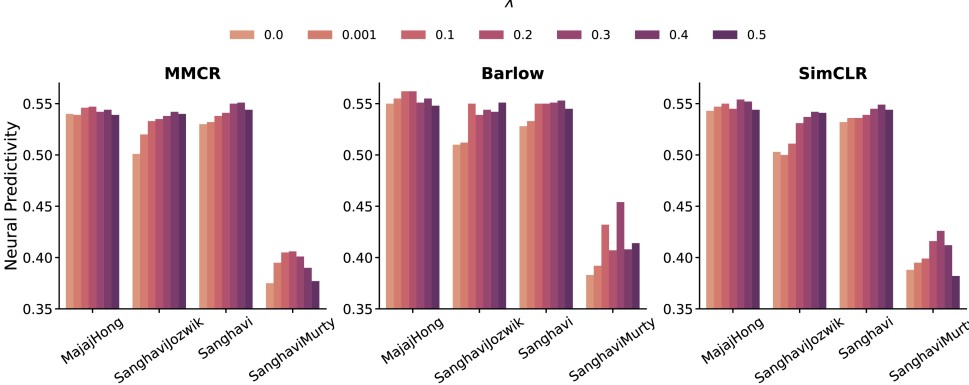

Figure 3: Brain-Score (noise-ceiled predictivity evaluated via ridge regression) for each value of $\lambda$ (different colored bars) for each IT dataset (groups of columns) and base objective functions (different figure panels). For all datasets and base objectives the invariant network ($\lambda = 0$, lightest bars) is outperformed by at least one equivariant network, and the spread in predictivity over values of $\lambda$ is significantly larger than the spread in predictivity over base objective functions for invariant networks.

such as training dataset [Tuckute et al., 2022, Conwell et al., 2023, Yerxa et al., 2024]. In contrast, encouraging equivariance produced much larger gains than choosing between different base invariant objectives: the range of predictivities over the sweep of $\lambda$ was around 4 times larger than the range of predictivities over objective functions for the invariant baseline. We further contextualize the scale of predictivity improvements in Fig 4 by comparing models to all public submissions on the BrainScore leaderboard; our equivariant training procedure improves performance of the already-strong invariant models to nearly state-of-the-art levels of IT predictivity. By training the most predictive model for 1000 epochs (rather than 100), we achieved 0.5355 mean fraction of explained variance, which makes this the top IT brain prediction model. To ensure that the observed alignment increases are not architecture specific, we trained a smaller set of models using different backbone architectures and observed similar trends when using both smaller and larger networks (see Appendix A.7 for details).

We quantified the correlation between our various representational measurements and the neural predictivity for each of the four electrophysiology datasets in Table 1. We observed that the only representational metric with a correlation greater than $0.4$ across all four neural datasets was the Spatial-Photometric distance, which is the metric most closely related to the factorization score described in [Lindsey and Issa, 2024]. While this previous study described a correlation between structured variability and neural predictivity measured from a large set of pre-trained models, our results demonstrate that explicitly encouraging such structures can improve alignment between artificial and biological representations. In addition to the previously described representational measurements, we also looked at the linear decoding of the hue modulation parameter in isolation. Hue modulation is one of 12 augmentation parameters that are linearly decoded in the parameter regression measures described in Section 3.2. We observed a strong correlation between neural predictivity and hue modulation, particularly with the Sanghavi-Jozwik dataset, which is the only response dataset that included color image stimuli (last column of Table 1).

### 3.4 Transfer Learning

Several previous studies that aim to reduce augmentation-invariance of self-supervised features have reported that the resulting representations generalize better to out-of-distribution classification datasets [Gupta et al., 2023, Xiao et al., 2020, Suau et al., 2023, Chavhan et al., 2022]. However most of these studies focused on using the smaller ImageNet-100 dataset for training, in one case reporting that the transfer learning gains diminish or disappear when using ImageNet-1k [Chavhan et al., 2022]. We tested our set of networks on 6 different downstream tasks and found limited evidence that the equivariant features confer an advantage in terms of out-of-distribution generalization when training on a sufficiently large and diverse dataset (see Table 2). To address this discrepancy with the literature we conducted additional experiments on networks trained using the ImageNet-100 dataset, and in this case observed improvement in generalization to diverse downstream tasks.

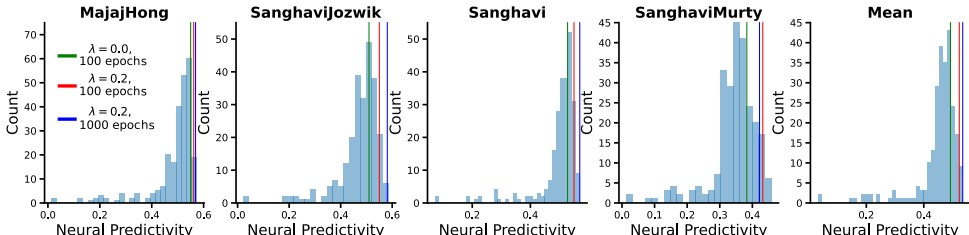

Figure 4: Histogram of neural predictivity scores for the 249 models on the public Brain-Score leaderboard at the time of testing, for each of the four considered IT datasets, as well as the mean over the four datasets. In each plot the vertical green line shows the score of the invariant Barlow Twins model, the red line shows the score for the equivariant Barlow Twins model with $\lambda = 0.2$, and the blue line shows a new model trained for 1000 epochs, also using Barlow Twins for the base loss and $\lambda = 0.2$.

Table 1: Absolute values of Pearson correlation coefficients ($R^2$) between various representational measurements and the neural predictivity across each of the four IT datasets. The correlation was measured over each value of $\lambda$ and base objective function for a total of 21 networks. Each column corresponds to a panel in Fig. 2, except for hue, which is the regression score obtained for the random hue modulation parameter in isolation.

| Neural Dataset | Aug-Aug | Aug-Centroid | Spatial-Photometric | Class-Class | Param Regression | Hue Regression |
|---|---|---|---|---|---|---|
| Majaj-Hong | 0.03 | 0.02 | 0.42 | 0.10 | 0.38 | 0.28 |
| Sanghavi-Jozwik | 0.86 | 0.7 | 0.83 | 0.77 | 0.91 | 0.91 |
| Sanghavi | 0.84 | 0.84 | 0.68 | 0.63 | 0.85 | 0.86 |
| Sanghavi-Murty | 0.33 | 0.41 | 0.42 | 0.14 | 0.56 | 0.48 |

It is also worth noting that CE-SSL trained networks do not outperform their invariant counterparts on in-distribution generalization (see A.2.1 and Fig. 5). This is not surprising in light of the fact that the suite of augmentations and architectures employed in SSL have been in some sense optimized by the community in order to improve performance on this task (by aligning the transformation invariance task with the standard in-distribution classification task). However, for out-of-distribution classification tasks where the task-alignment is worse, the equivariance task could mitigate this mismatch. A concrete example is the Flowers-102 dataset, where the color of petals is a much stronger predictor of class than color is in, say, the ImageNet-1k dataset (so the color insensitivity induced by the standard augmentations could be detrimental). For this dataset we do see marginal improvements, but note that the improvements are much more pronounced when pretraining on smaller datasets (ImageNet-100). There are at least 2 possible explanations for this: (1) for ImageNet-1k pretraining the performance of the networks is already quite high, and the task is saturated, or (2) there is a more fundamental reason that the improvements in transfer learning induced by equivariance decrease as the size and diversity of the pretraining dataset grows. Future work could explicitly disambiguate between these hypotheses to determine why the benefits of transformation-related variability for out of distribution generalization are outweighed by the gains of scaling the dataset. Furthermore this result shows that the increased neural predictivity we observe in ImageNet-1k trained networks cannot be explained by a need to perform better on a variety of invariant-classification tasks.

## 4 Relationship to Existing Augmentation-Sensitive SSL Methods

A key feature that differentiates our approach is that it encourages equivariant structure without explicit access to augmentation parameters. This is enabled by the "paired augmentation" data generation procedure, and to the best of our knowledge CARE [Gupta et al., 2023] is the only existing work that shares this feature. Our method has two advantages over CARE: (1) in CARE the equivariance loss is applied in the same space as the invariance constraint, and because there are no "negative equivariant pairs" in the CARE framework, learning an invariant representation would perfectly satisfy the equivariant constraint; and (2) in CARE the standard augmentation pipeline is

Table 2: Frozen-Linear Evaluation for invariant and equivariant trained networks on 6 different downstream datasets: Cifar-10/100 [Krizhevsky et al., 2009], Oxford-Pets Parkhi et al. [2012], Describable Textures Database [Cimpoi et al., 2014], Flowers-102 [Nilsback and Zisserman, 2008], and Food-101 [Bossard et al., 2014]. We closely follow the evaluation procedure from [Lee et al., 2021] (see Appendix A.4 for details) and report top1 accuracy for each objective/dataset. In all cases we report the mean over 5 runs of the evaluation procedure, we observed very little variability (maximum of .2%, over all evaluations, we report the standard deviation over runs in Appendix A.4. The equivariant networks are denoted by prepending a "CE" before the objective and were trained using $\lambda = 0.1$, which enabled a substantial amount of structure variability without significantly impacting frozen-linear classification on the SSL training dataset (see Appendix A.2). For ImageNet-1k trained networks out of distribution performance decreased for most evaluations, while for ImageNet-100 trained networks performance was improved in 15 of 18 cases.

| *ImageNet-100 Training* | | | | | | |
| Objective | Cifar-10 | Cifar-100 | Pets | DTD | Flowers-102 | Food-101 |
| --- | --- | --- | --- | --- | --- | --- |
| MMCR | 84.3 | 63.3 | 67.0 | 66.1 | 83.1 | 60.9 |
| Barlow | 87.7 | 68.7 | 74.8 | 67.0 | 88.3 | 63.4 |
| SimCLR | 87.8 | 68.8 | 74.3 | 66.6 | 88.5 | 64.8 |
| CE-MMCR | 87.3 | 69.4 | 68.9 | 65.7 | 87.5 | 64.1 |
| CE-Barlow | 88.0 | 69.1 | 73.6 | 67.3 | 89.5 | 65.5 |
| CE-SimCLR | 87.9 | 68.2 | 72.6 | 67.5 | 88.6 | 65.2 |
| *ImageNet-1k Training* | | | | | | |
| Objective | Cifar-10 | Cifar-100 | Pets | DTD | Flowers-102 | Food-101 |
| MMCR | 92.2 | 76.9 | 85.3 | 75.4 | 93.9 | 73.8 |
| Barlow | 91.8 | 75.8 | 86.5 | 73.0 | 93.8 | 72.2 |
| SimCLR | 91.8 | 74.7 | 85.1 | 74.5 | 92.7 | 70.5 |
| CE-MMCR | 92.2 | 76.6 | 84.3 | 75.7 | 94.0 | 73.8 |
| CE-Barlow | 91.8 | 75.3 | 85.7 | 75.6 | 94.2 | 73.0 |
| CE-SimCLR | 91.0 | 73.6 | 82.3 | 73.9 | 92.2 | 70.5 |

used to optimize the base $\mathcal{L}_{iSSL}$ loss and paired augmentation are used in parallel to optimize the equivariance term (so an increased number of passes through the network is necessary relative to standard training).

In EquiMod [Devillers and Lefort, 2023] the projector network is conditioned on the augmentation parameters by appending the parameters to the output of $f$. The authors theorize that knowledge of the augmentation parameters could allow the projector network to better extract an invariant subspace tailored to each transformation, thus allowing for more structured variability in the representation space. Alternatively, the projector network could simply ignore the augmentation parameters, resulting in a structure that is identical to invariant SSL. In practice [Garrido et al., 2023b] have found this to be the case. Split-Invariant-Equivariant (SIE) and Amortised Invariance (AI) learning [Garrido et al., 2023b] each improve on this principle by using a hypernetwork approach: a separate network takes as inputs the augmentation parameters and outputs the parameters of either $g$ or both $f$ and $g$ respectively. While the collapse issue of EquiMod is avoided, this comes at the expense of significantly complicating the network computation, and in the case of AI introduces new parameters that need to be tuned for every downstream task (when augmentation information is not available). Still other methods supplement the standard invariant SSL loss with an auxillary term that involves predicting the parameters of the input transformation [Lee et al., 2021, Dangovski et al., 2021]. The relationship of our method to these is analogous to the relationship of transformation-invariant self-supervised learning to supervised classification.

# 5 Discussion

We've developed a new self-supervised objective that explicitly encourages structured variability in networks, and demonstrated that it can produce increased alignment with responses of neurons in primate visual area IT. While we are not the first to incorporate a notion of equivariance to self-supervised learning, our method improves on existing work in several ways: it require no extra passes through the network relative to invariance-based learning, it encourages diversity in the representation of transformation-related information by leveraging advances in invariance-based learning, and it does not rely on supervised access to transformation parameters. The parsimony of our approach (applying the same objective to both outputs of individual images and to displacements between similarly transformed images) allows our technique to be easily adapted to other settings such as temporal self-supervised learning (discussed below). Although in this work we focused on the visual domain, similar equivariant and invariant objectives could be investigated for other domains such as audio and langauge representation learning.

Our approach induced several interesting features in the learned representations: transformation variability is shared across base images and factorized with respect to variability over base images, the variability induced by distinct types of transformations are factorized from each other, there is increased alignment between class manifolds, and transformation related information is linearly encoded. Some of these properties are closely related to the imposed objective and some are emergent. We also confirmed that several of these representational properties are correlated with increased neural predictivity. Future work can extend these correlative observations to better understand how increasing transformation sensitivity improves neural alignment. For example, one could analyze the residuals of predicted neural firing rates of distinct models to determine how "overlapping" the variance predicted by each is (or alternatively, attempt to fit the residual variance of one model with another). Such analyses are becoming more feasible with the collection and release of larger scale datasets of neural responses to natural images (e.g., [Madan et al., 2024]). We view this result as demonstrating the promise of incorporating knowledge gained from experimental observations and large scale comparative studies into optimization procedures to produce better models.

Although our experimnents reveal both induced and emergent benefits, the inclusion of an additive equivariance term in the objective does lead to fewer guarantees regarding the learned structure. For example in schemes where the output transformations ($T_\rho$'s) are explicitly represented or learned, the resulting representation is "steerable" by default. It is of interest to investigate whether the output transformations could be reliably recovered from our learned representations. Additionally it would be interesting to consider other types of output transformations (CARE [Gupta et al., 2023] focuses on orthogonal transformations, and in the case where output transformations are learned they can be computed with nonlinear neural networks).

Finally, it is of interest to explore the use of more ecologically relevant sources of training data, e.g., by replacing synthetically transformed views of images with temporally adjacent frames of natural videos. This approach is particularly appealing from the perspective of biological plausibility, as the pairing of such training examples is readily available from natural visual experience. Several recent publications have shown that such a strategy can produce representations with competitive neural predictivity and performance on computer vision tasks [Zhuang et al., 2021, Parthasarathy et al., 2023, Venkataramanan et al., 2023]. In this context, the typical invariance loss can be thought of as incentivizing representational slowness [Földiák, 1991, Wiskott and Sejnowski, 2002]. The equivariance mechanism described in this work could be implemented by applying the same invariance-based loss function to the first temporal derivative of the responses, i.e. by encouraging the displacement between successive pairs of frames to be constant. Such a temporal-equivariance objective would incentivize representational straightness, which has been used to describe features of both human perception and neural activity in the ventral stream [Hénaff et al., 2021, 2019]. Straightness in artificial representations has been found to be correlated with both neural predictivity and adversarial robustness [Lindsey and Issa, 2024, Harrington et al., 2022, Niu et al., 2024]. These connections provide an array of promising research directions.

## Acknowledgments and Disclosure of Funding

This work was funded by the Center for Computational Neuroscience at the Flatiron Institute of the Simons Foundation. S.C. is supported by the Klingenstein-Simons Award, a Sloan Research Fellowship, NIH award R01DA059220, and the Samsung Advanced Institute of Technology (under the project "Next Generation Deep Learning: From Pattern Recognition to AI"). All experiments were performed on the Flatiron Institute's high-performance computing cluster.

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

# A Appendix

## A.1 Reproducibility

All code used for pretraining, evaluation, and analyses, will be made available in a public github repository upon publication.

## A.2 Additional Pretraining Details

Here we report some additional hyperparameters not included in the main text.

**Optimization:** For all experiments we trained for 100 epochs using a batch size of 2048 and used the LARS optimizer [You et al., 2017] with weight decay of 1e-6 and momentum of 0.9. Note that the $\mathcal{L}_{CE-SSL}$ loss is evaluated on pairs of augmented views and thus had an effective batch size of 1024. We use a base learning rate of 4.8 and a learning rate schedule consisting of linear warm-up for the first 10 epochs followed by cosine decay throughout training.

**Projector Architectures** Each trained network uses two projectors with matching architectures. For Barlow Twins we used the architecture proposed in the original work [Zbontar et al., 2021] (3 layer MLP with hidden layer and output layer widths of 8192). For MMCR we also used a 3 layer MLP with 8192 hidden width but 512 output units (also in line with the original work [Yerxa et al., 2024]). For SimCLR we used the same projector architecture as MMCR, which is larger than the MLP described originally because subsequent work [Garrido et al., 2023a] has found that SimCLR benefits from a more expressive projector.

### A.2.1 ImageNet-1k

For Barlow Twins we set the $\lambda_{BT}$, which balances the on and off diagonal loss terms, hyperparameter to $5e-3$. For SimCLR we used a temperature of $\tau = 0.15$.

### A.2.2 ImageNet-100

Besides the change of dataset, the only hyperparameter change in this setting is that we increased the number of pretraining epochs from 100 to 200 to be more in line with previous work.

## A.3 Online-Linear Evaluation for the Pretraining Dataset

Because frozen-linear evaluation on large datasets is computationally intensive we instead opt for online-linear classification. During pretraining the representation network outputs are detached from the gradient propogation graph and fed through a linear layer that is optimized with the standard supervised cross entropy loss. Previous work has shown that online-evaluation is very strongly correlated with frozen-linear evaluation and incurs only a minimal cost on top of self-supervised pretraining. We report the accuracies for ImageNet-1k trained networks in Fig. 5 and the smaller set of ImageNet-100 trained models in Table 3.

| Model | Accuracy |
|-----------|----------|
| MMCR | 79.0% |
| Barlow | 81.4% |
| SimCLR | 83.5% |
| CE-MMCR | 79.6% |
| CE-Barlow | 81.6% |
| CE-SimCLR | 82.5% |

Table 3: In distribution accuracy on the validation set of ImageNet-100 evaluated by online-linear classification.

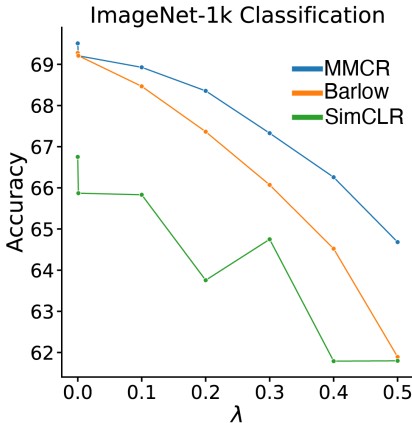

Figure 5: In distribution accuracy on the validation set of ImageNet-1k evaluated by online-linear classification, for each objective and as a function of $\lambda$.

## A.4 Transfer Learning Evaluation Procedure

We closely follow the evaluation procedure from [Lee et al., 2021], we repeat the details here for completeness. First images are resized such that the shortest edge is 224 pixels, then center cropped to 224x224 resolution. Then features are extracted from train, validation, and test splits of each dataset. L-BFGS is used to optimize the standard cross entropy loss with $\mathcal{L}_2$ regularization, the value of the ridge parameter is swept over selected via performance on the validation set. Subsequently the linear classifier is retrained using both the train and validation sets, and we report the final accuracy on the held out test set.

This classification procedure was run 5 times with different random initializations, we reported the mean performance in 3.4, and report the standard deviation over runs below.

Table 4: Standard deviation of top 1 accuracies over 5 independent runs of the transfer learning evaluation procedure.

| *ImageNet-100 Training* | | | | | | |
|---|---|---|---|---|---|---|
| Objective | Cifar-10 | Cifar-100 | Pets | DTD | Flowers-102 | Food-101 |
| MMCR | 1e-2 | 7e-3 | 4e-2 | 1e-5 | 7e-2 | 2e-1 |
| Barlow | 1e-4 | 2e-4 | 2e-2 | 3e-1 | 3e-2 | 9e-3 |
| SimCLR | 1e-2 | 7e-3 | 1e-2 | 4e-2 | 7e-2 | 5e-3 |
| CE-MMCR | 1e-2 | 2e-2 | 4e-2 | 3e-2 | 1e-1 | 1e-2 |
| CE-Barlow | 5e-2 | 2e-2 | 3e-2 | 3e-2 | 8e-2 | 1e-2 |
| CE-SimCLR | 5e-3 | 2e-2 | 2e-2 | 2e-2 | 5e-2 | 1e-2 |
| *ImageNet-1k Training* | | | | | | |
| Objective | Cifar-10 | Cifar-100 | Pets | DTD | Flowers-102 | Food-101 |
| MMCR | 1e-2 | 2e-2 | 8e-2 | 4e-2 | 2e-2 | 7e-3 |
| Barlow | 1e-2 | 1e-1 | 1e-1 | 4e-2 | 3e-2 | 3e-3 |
| SimCLR | 2e-2 | 7e-3 | 1e-4 | 4e-2 | 2e-2 | 8e-3 |
| CE-MMCR | 9e-3 | 2e-2 | 2e-1 | 2e-1 | 6e-2 | 2e-2 |
| CE-Barlow | 1e-5 | 1e-2 | 9e-2 | 1e-5 | 5e-2 | 9e-2 |
| CE-SimCLR | 1e-2 | 1e-2 | 2e-2 | 2e-2 | 7e-2 | 8e-3 |

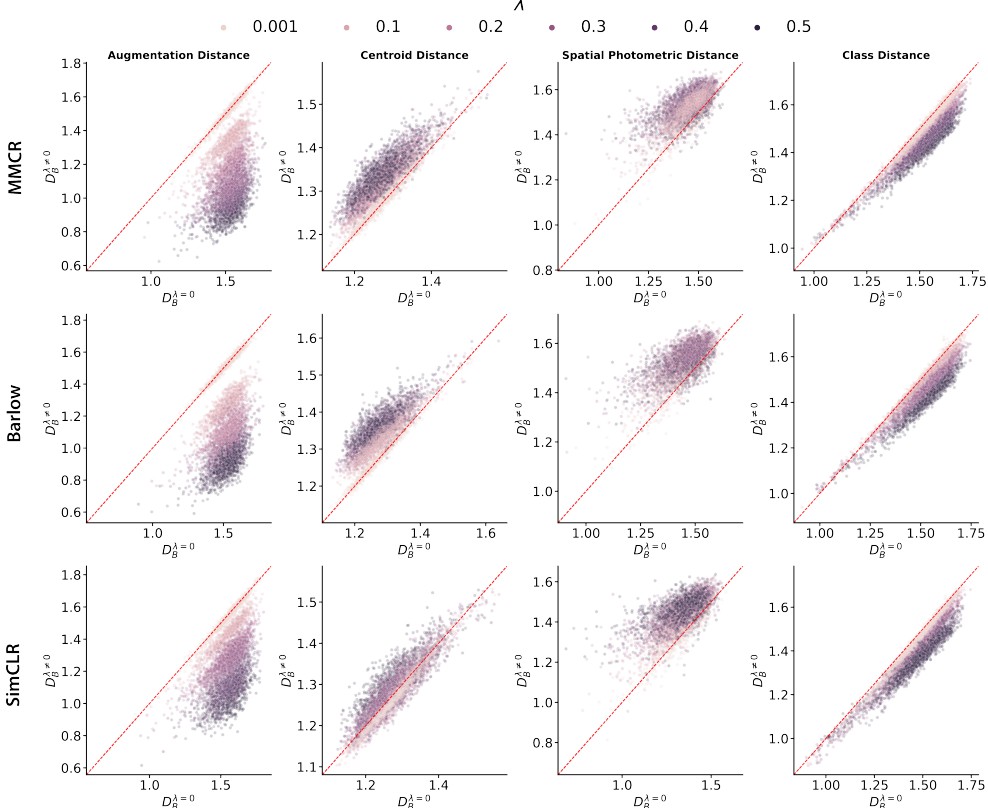

Figure 6: Joint distributions of invariant and equivariant networks (increasing $\lambda$ increases the importance of the equivariant loss) for all of the Bures metric comparisons detailed in 3.2. The mean value curves in Fig. 2 are generated by taking the mean of $x - y$ for each of these plots, separately for each objective and value of $\lambda$. The confidence intervals are estimated from the distribution of $x - y$ values as well. Columns depict Bures distances between covariances estimated over different sources of variability, and columns index different base invariant objective functions.

## A.5 Additional Details for Representational Analyses

Below we depict the joint distributions of the distances described in 3.2. For each setting (unique objective function and value of $\lambda$ there are 800 unique measurements (i.e. points of a unique hue on an individual plot). Meaning, for example, there are 800 random pairs of augmentation manifolds compared in both invariant representation space and equivariant representation space for each equivariant network in the left most column. We summarize describe the sources of variability over which covariance matrices are estimated and the variables over which the expected Bures distance is calculated for the experiments in Fig. 2 A-D.

## A.6 Out-of-distribution Equivariance

We aim to test whether learning equivariances using weak augmentations induces structured variability in response to stronger augmentations (the extent to which learned equivariances generalize beyond the range of transformations seen during training). We trained models using the Barlow Twins objective and the same sweep over values of $\lambda$ using "weak" augmentations of (1) double the minimum crop size, (2) half the maximum value of color jittering, and (3) half the maximum size of Gaussian blurring kernel. We then repeat the parameter decoding experiments from the main paper Fig. 2E on these weak augmentations (left panel Fig. 7), and on the non-overlapping part of the parameter space between the weak and strong augmentation distributions, i.e only for augmentations whose parameters are in distribution for the models trained as in the main text but out of distribution for the new models trained using weaker augmentations (right panel Fig. 7). In the first panel,

Table 5: Table defining the sources of variability producing covariance matrices and the random variables that the expected Bures distance is computed over for the experiments in Fig. 2.

| Column | Source of Variability $C_1$ | Source of Variability $C_2$ | Ensemble for estimating expectation |
|---|---|---|---|
| Augmentation-Augmentation Distance (A) | Augmentations of single image. | Augmentations of single image. | Random pairs of distinct images |
| Augmentation-Centroid Distance (B) | Augmentation manifold centroids over all images | Augmentations of single image. | Random images (those used to compute $C_2$) |
| Spatial-Photometric Distance (C) | Photometric augmentations (of a single image) after averaging over many random crops. | Random crops (of a single image) after averaging over many photometric augmentations. | Unique base images. |
| Class-Class Distance (D) | (Unaugmented) exemplars from one class | (Unaugmented) exemplars from one class | Random pairs of distinct classes |

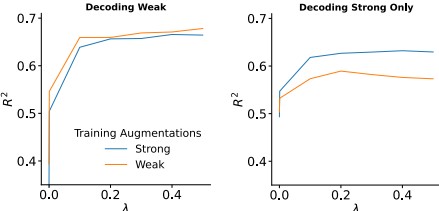

Figure 7: Augmentation parameter decoding performance on held-out test images for networks trained using either strong or weak transformations. In the left panel we plot the decoding performance for weak transformations only, and in the right panel the performance for strong transformations only (which are not seen by the weak-trained networks during pretraining).

we can see that the best parameter decoding performance occurs when the pretraining distribution of transformations is matched to the evaluation transformations (i.e. weak-trained models slightly outperform the strong-trained models at decoding the parameters of weak transformations). In the right panel we see that models trained with strong augmentations have higher decoding performance, but models trained only on weak augmentations still demonstrate significantly increased ability to linearly decode strong augmentation parameters relative to the invariant trained models ($\lambda = 0$ models), indicating a degree of generalization in the learned equivariances.

## A.7 Different Backbone Architectures

To verify that the effect of equivariance neural predictivity observed in the main text is not limited to a specific choice of architecture, we trained invariant ($\lambda = 0.0$) and equivariant ($\lambda \in [0.1, 0.2]$) networks using $\mathcal{L}_{BT}$ with smaller (ResNet-34) and larger (ResNet-50) backbones. As shown in Fig. 8, we observe the same trend across different architectures: contrastive-equivariant training increases alignment to area IT as measured by linear predictivity.

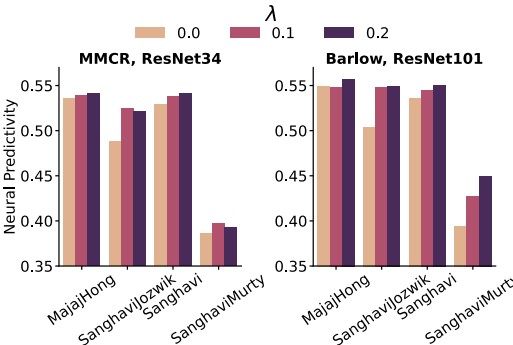

Figure 8: Neural predictivity results for models trained using the MMCR base loss and a ResNet-34 backbone (left panel), and the Barlow Twins base loss and a ResNet-101 backbone (left panel). We see similar trends in terms of increased neural predictivity for equivariant trained models as observed using the ResNet-50 backbone in the main text.

## A.8  Compute Resources

All pretraining runs used 8 A100 Nvidia GPUs with 40GB of memory each. In our setting pretraining run times were around 15 hours, and we note that CE-SSL training generally increased training time by approximately 10% relative to standard self-supervised training. Subsequent evaluations ran on a single A100.

## A.9  Limitations

We discuss some limitations not addressed in the discussion here. Our current training setup requires selection of the hyperparameter $\lambda$ to balance between the equivariant and invariant loss functions. Future work could investigate methods to balance the two losses without explicitly training an individual network for each choice of $\lambda$. It is worth noting that some invariant SSL methods may be more or less sensitive to this additional hyperparameter. For example, some of the non-monotonicity of SimCLR curves in Fig. 2 may suggest that SimCLR representations are more difficult to smoothly shape within the CE-SSL paradigm.

Additionally, computational limitations prevent us from doing an extensive architecture search over the two projector networks employed in contrastive equivariant training. In addition to varying the depth and width of each projector, it would be of interest to "split" the representation space and have each projector operate on a subset of dimensions as input (as in SIE [Garrido et al., 2023b]). Additionally computational limitations prevented us from extensively evaluating the variability in neural predictiivity over independent runs of contrastive equivariant training (the BrainScore framework recently stopped providing estimates of the error of neural predicitivity, but in previous studies the reported error for IT predicitivity was $\approx$ 3e-3, which is small relative to the variability we observed across the parameter of interest $\lambda$). Pilot experiments indicated to us that the variability over training runs was small relative to the variability over values of lambda (we trained two invariant models and two with $\lambda = 0.1$ to get a rough estimate of this, in both cases we kept the more predictive model for inclusion in all analyses that appear in this paper).

## A.10  Broader Impacts

In this work we propose one strategy for inducing increased alignment between artificial and biological visual representations. Better understanding the computational principles underlying visual representations has the potential to benefit the quality of computer vision applications, to offer insights into the structure of the primate visual system, and to improve clinical treatment of disorders related to visual perception.

