# OpenReview forum: "Contrastive-Equivariant Self-Supervised Learning Improves Alignment with Primate Visual Area IT"
_NeurIPS.cc/2024/Conference — NeurIPS 2024 poster_

### Official Review · Reviewer_Fzg3 · 2024-07-12

**Soundness:** 3
**Presentation:** 4
**Contribution:** 3
**Rating:** 7
**Confidence:** 3

**Summary:**

This paper presents an equivariant learning framework that modifies standard invariant-based self-supervised learning methods by integrating differences between object classes (Loss_iSSL) and the changes observed in images before and after identical transformations (Loss_CE-SSL). This adaptation enables the model to effectively incorporate transformation-related variability without necessitating additional transformation parameters or extensive modifications to the training protocol. The authors then demonstrate that this approach systematically enhances the model's capacity to predict neural activity in the inferotemporal (IT) cortex, revealing that optimizing for structured variability significantly boosts the accuracy of predicting cortical responses to natural images.

**Strengths:**

1. The article is clearly articulated and well-defined.
2. It introduces an effective self-supervised objective function for implementing equivariance, which can be applied across various models including SimCLR, MMCR, and Barlow Twins.
3. The paper conducts numerous experiments to analyze the impact of this training method on representations. It compares the relationship between different intensities of equivariance error (Loss_CE-SSL) and monkey IT neuron activities, as well as the model's performance in transfer learning tasks.

**Weaknesses:**

1. The model's performance in transfer learning tasks, as shown in Table 2, does not demonstrate a significant improvement; in many cases, it merely maintains accuracy levels similar to those before implementation.
2. While the paper analyzes representations based on different values of $\lamda$, these analyses seem to offer little help in predicting neuronal activity or improving transfer learning. The proposed "tradeoff between invariance and structured variability" does not appear to manifest significantly.
3. The equivariance learned by the method may be limited by the types and varieties of transformations applied to the images, an aspect the paper does not analyze or discuss.

**Questions:**

1. The SimCLR curve in the lowest part of Figure 2 appears more jagged compared to the other two models. Could this indicate that SimCLR is less suitable for incorporating equivariance?
2. When predicting IT neuron activity, how can we interpret the variations caused by different $\lamda$ values across datasets?
3. In the transfer learning results, why does the addition of an equivariance loss seem to have minimal impact on the outcomes?
4. Is the equivariance learned by this method limited by the types and varieties of transformations applied to the images? For instance, if the model only learns transformations involving rotations between 0-30 degrees, can it represent equivariance for a 60-degree rotation? Furthermore, can the method effectively learn more complex, non-linear transformations, and can learning multiple transformations enable the model to represent combinations of these transformations equivariantly?
5. There appears to be a typographical error in line 225: “($\lamda$ = 0) for all four datasets (Fig. 3.3),” as the figure number seems incorrect.

**Limitations:**

Yes, the paper concludes with a discussion on some limitations of the model and directions for future research.

---

> ### Author Rebuttal · Authors · 2024-08-07
>
> Thank you for your constructive review of our submission. We respond to the questions and limitations point-by-point below:
>
> - Performance on Downstream Classification Tasks (W1, Q3): Please see our general response (Point 3) for our more detailed thoughts on why equivariant training on a significantly large dataset offers little in the way of improvements on downstream classification tasks. In short, our results on the ImageNet-100 training vs. ImageNet-1000 training suggest that the benefits induced by equivariant training “overlap” with those that come from using a large dataset (Line 252 of the submission). With this result, we aim to draw attention to the underappreciated fact that equivariant training seems to be more valuable when the pretraining dataset is smaller.
>
> - Lack of a tradeoff between invariant and equivariant terms (W2): We think there might have been some confusion surrounding our use of the word “tradeoff”; when we refer to a “tradeoff between invariance and structured variability” (Line 54) we are referencing qualities of the artificial representations, not necessarily proposing that IT neurons are explicitly striking some balance between invariance and equivariance. In the artificial representations such a tradeoff is clear, for example, as the importance of the equivariance loss increases the amount of linearly accessible information about transformations applied to the input (Figure 2E), but this comes at the cost of a modest decrease in performance on classification on the in-distribution ImageNet-1k dataset (Figure 4). Furthermore while the improvements in neural predictivity are small in absolute terms, this is common for linear-regression based comparisons, and our improvement does result in a substantial increase in terms of the relative ranking of models on the IT Brain–Score leaderboard (see our general response and Figure R2 for more details).
>
>     If we have misinterpreted this we are more than happy to engage further during the discussion period! We also will remove the reference to a “tradeoff” in line 58 with the following modification, to help avoid confusion:
>     “We explore the impact of including an equivariant loss for predicting neural activity in IT,”
>
> - Dependence on the input transformations (W3, Q4): We fully agree that this is an important point and could be better addressed. Thank you in particular for suggesting we experiment with testing how well the learned equivariances can generalize to unseen (stronger, but similarly typed) transformations. As a step towards answering this question we trained models using the same suite of augmentations but with a uniformly reduced strength, and repeated the parameter decoding experiments from Figure 2. These preliminary results suggest there is a degree of generalization to stronger transformations, see our general response and Figure R1 of the attached PDF for more details. We are currently working on the experiment you described involving rotations, and hope to have additional results to report during the discussion period.
>
>     Considering other more complicated sets of input transformations than those commonly used in SSL is also an interesting direction for further research that we can highlight more directly. We feel the most important and ecologically relevant set of input transformations are those that mimic the visual experience of animals in the world, and as we note in the discussion we believe this is a very promising direction for future work!
>
> - Is SimCLR less suited to incorporating equivariances (Q1): Our results do suggest that SimCLR may indeed be  less amenable to smoothly shaping the representation via an equivariance loss. However another  hypothesis is that these nonsmooth curves arise from the fact that SimCLR is in general more sensitive to various hyperparameter choices, and the one introduced by our equivariance loss is not an exception. For example, it is well known that SimCLR is more sensitive than other self-supervised learning methods to batch size [1]. To definitively determine whether this is a fundamental difference between SimCLR and the other objective functions would require large sweeps over a variety of hyperparamters (batch size, learning rate, etc.), and compute limitations prevent us from conducting such experiments. We will expand Appendix A.7: Limitations, to include this discussion.
>
> - Interpreting the impact of the equivariance loss on neural predictivity (Q2): We agree that gaining better understandings and intuitive interpretations of how different features of learned representations lead to different levels of neural predictivity is an important goal for both our work and the field at large. We attempt to draw conclusions by analyzing how the representational analyses from Section 3.2 correlate with neural predictivity in Table 1. However we agree that there is more work to be done in terms of interpretation. For example one could examining the structure of the residuals and mapping weights obtained in the Model-to-Brain regression problem [2]. We could also begin to answer some of these questions using new and open source datasets of neural measurements in Macaque IT [3] and will highlight the importance of such investigations in the revised manuscript.
>
> - Typos (Q5): Thank you for drawing our attention to this mistake, we will be sure to correct the figure reference in the revised manuscript.
>
> [1] Zbontar, Jure, et al. "Barlow twins: Self-supervised learning via redundancy reduction." International conference on machine learning. PMLR, 2021.
>
> [2] Canatar, Abdulkadir, et al. "A spectral theory of neural prediction and alignment." Advances in Neural Information Processing Systems 36 (2024).
>
> [3] Madan, Spandan, et al. "Benchmarking Out-of-Distribution Generalization Capabilities of DNN-based Encoding Models for the Ventral Visual Cortex." arXiv preprint arXiv:2406.16935 (2024).

---

> > ### Comment · Reviewer_Fzg3 · 2024-08-09
> >
> > Thank you for your response. I really like the results presented in Figure R1 and am looking forward to seeing the attempts involving rotations. These results have convinced me well, and I will increase my score.

---

> > > ### Author Response · Authors · 2024-08-13
> > > **Response**
> > >
> > > Thank you for taking the time to consider our rebuttal experiments. We are hard at work on the rotation case! Preliminary results suggest a trend similar to that of Figure R1.

---

### Official Review · Reviewer_AoJ5 · 2024-07-15

**Soundness:** 3
**Presentation:** 2
**Contribution:** 3
**Rating:** 6
**Confidence:** 4

**Summary:**

Summary:

The authors propose a novel approach to self-supervised learning (SSL) by incorporating contrastive equivariant training into existing successful SSL methods based on creating invariance to input transformations. The authors observe that such an incorporation of equivariant contrastive learning produces the following downstream benefits: 1) Improved representational similarity to biological visual representations, and 2) Enhanced transfer learning performance on downstream tasks. Both these findings have been evaluated via experiments performed using the BrainScore benchmark and by computing linear probe classification accuracy of various SSL training methods on downstream image classification tasks.

**Strengths:**

Strengths:
- The proposed contrastive equivariant SSL (CE-SSL) is a straightforward yet effective extension of existing invariance-based SSL methods. This enhancement has broad potential extending beyond this submission, as CE-SSL-trained encoders may excel in downstream tasks that rely on equivariance, such as image segmentation.
- The experimental results demonstrate a significant improvement in two key areas: 1) Decoding performance of macaque IT neural recordings as validated by their model performance on BrainScore, and 2) Transfer learning to downstream image classification problems as validated by their linear probe classification experiments presented in Table 2. Evaluation performed broadly over 3 different SSL techniques over multiple $\lambda$ parameters enhances the technical soundness of this work.
- The authors provide a clear discussion of the proposed work's relationship to prior art in equivariant self-supervised learning, situating their contributions within the broader context of the field.

**Weaknesses:**

- Overall, the improvements from using CE-SSL in the BrainScore performance is only quite marginal, hence raising the question of how impactful the current work will be in the broader scheme of methods to enhance neural predictivity.
- The writing quality of this work could be further improved. Particularly, Section 2 is currently written in a convoluted manner in my opinion and could be further improved to enhance readability. There are other minor writing issues, for e.g.:
    - Lines 145-146 and equation 2 are in conflict with each other. Throughout the paper, it looks like $\lambda=0$ refers to the invariant case. In that case, the invariant term of Eqn. 2 should be weighted by $(1-\lambda)$. Is that correct? Currently the invariant term is weighted by $\lambda$ in which case $\lambda=0$ would correspond to training only with CE-SSL.
    - Line 113, I believe the authors intended to refer to Figure 1 but they incorrectly refer to Figure 2 in the submission
    - Citation issue in Line 86, Zbontar et al. [2021].
    - It would help to use different icons to represent $f$ and $g_{inv}$ or  $g_{equi}$ as there are significant structural differences between the backbone and projector networks.

- In Figure 4, as the CE-SSL loss contribution increases, the ImageNet classification accuracy seems to be dropping. This is an issue that needs to be explicitly discussed more clearly.

**Questions:**

Please refer to my review above.

**Limitations:**

Limitations are discussed adequately in this submission.

---

> ### Author Rebuttal · Authors · 2024-08-07
>
> Thank you for your thoughtful assessment of our paper. Find below is our point-by-point response to the weaknesses and questions raised:
> - Marginal Effect on Neural Predictivity: Please see our general response Point 1 and Figure R2 in the rebuttal pdf which directly address this concern about the marginal effect on IT predictions. Specifically, the changes observed are substantial in terms of the relative ranking compared to other models on this benchmark.
> - Writing/Clarity Issues: You are correct, there is a misleading typo in Eqn. 2 and the weightings of the two terms are reversed. Thank you for pointing out this mistake, we will correct it in the revised manuscript to  $L_{\rm overall} =  (1 -  \lambda) L_{iSSL} + \lambda L_{CE-SSL}$. Similarly the reference on line 113 should indeed be to Figure 1. Finally, we will also correct the citation error on line 86 and change the icon for projector networks in Figure 1 to better indicate the transition from a convolutional backbone to an MLP projector.
>
> - Performance on ImageNet-1k Classification: Please see our general response Point 3, which provides reasoning for why downstream accuracy on ImageNet-1k degrades as we increase the importance of the equivariance loss. We agree that this issue could be better highlighted in the main text and propose to include language similar to that in our general response in the final version of the paper.

---

### Official Review · Reviewer_d1rT · 2024-07-15

**Soundness:** 3
**Presentation:** 3
**Contribution:** 2
**Rating:** 8
**Confidence:** 4

**Summary:**

In their paper ‘Contrastive-Equivariant Self-Supervised Learning Improves Alignment with Primate Visual Area IT’ the authors construct a new kind of contrastive-equivariant loss to optimise ResNet-50 architectures. Their new loss is based on the idea of using both self-supervised learning via object representations invariant across transformations of an image, and equivariant representations of transformation across images. They show that their new loss results in improved alignment to brain data recorded in IT. In a series of additional investigations, they show which part of the achieved representations is most likely causing the alignment to data. They also show that their method shows similar generalization advantages like prior alignment techniques but that this improvement is actually easily compensated for by training on a larger dataset.

**Strengths:**

I believe that the new loss developed in this paper is a very interesting development, routed in ideas which have been discussed in the field. The paper’s descriptions of ideas and results are overall nicely written and give a very balanced account of the investigations. The improved performance on the IT matching score is interesting, especially in light of current debated about self-supervised learning in the brain.

**Weaknesses:**

I believe that the two weaknesses, one of experimental nature and one on the writing:

Experimental –
The main goal of authors was to achieve higher alignment to brain data, but I found the comparison to other methods a bit thin. Authors say their method currently ranks 10th but it would be helpful to at least have a feeling for how far off the method is from 1st place. Many readers will not know whether the differences in the top 10 are in the magnitude of full percentage points or more / less. As such, I would suggest to perhaps at least provide the performance of the 11th and 9th, and 1st. This should be easy to address.
I also wonder how ‘unique’ the achieved explainability of their method is, i.e. authors contrast their method with mostly task-optimised networks and would they expect that adding their method to task optimised networks would result in an overall best performant network or is the explainability captured by their method also the variance captured by task optimisation so that combining them does not promise any further improvements? I can see that from a biological plausibility perspective the newly proposed technique is still nicer, but if the classification loss achieves learning even more brain-like presentations which also contain a similar trade-off between invariance and equivariance, then perhaps there is multiple ways for the brain to learn such representations and the loss presented here would not achieve any unique advancements.

Writing –
I found the section ‘3.2 Representational Analyses’ somewhat difficult to follow – I appreciate that authors with Figure 2 tried to provide visualisations of manifolds for different scenarios but perhaps it would be more helpful to visualise which cases are actually compared for which distance? Perhaps this could be done in the form of table in the appendix, if otherwise authors run out of space? From the text and current figures I struggled to exactly get which images / augmentations etc go into which set to construct the manifolds. Also, I believe the line plots at the bottom of Figure 2 would be significantly easier to understand if authors would use something like ‘Strength of equivariance loss (Lambda)’.

**Questions:**

Apart from the suggestions mentioned in weaknesses, I have some minor comments:
-	Some of the in-line references seem to be lacking the brackets in the proper locations, I noticed this in the references in line 86 and 227, where no overall bracket is around the reference but only around the year
-	In line 161 & 162 authors say ‘for each equivariant network’ and I am somewhat confused why there is a plural here? Does this mean the projection relating to each specific transformation?
-	In line 113 you mention a grey inset in Figure 2 but I am not sure what you are referring to with this?

**Limitations:**

The limitations discussed in the discussion and appendix seem appropriate.

---

> ### Author Rebuttal · Authors · 2024-08-07
>
> Thank you for your constructive assessment of this submission. Below is our point-by-point response to the weaknesses and questions raised:
> - Lack of baselines/context for brain data: We agree that the contribution is clarified by providing more context for the predictivities for a range of models. See our general response Point 1 and the Figure R2 in the attached PDF for our plan to do so.
>
> - Uniqueness of Improvements: This is indeed a relevant consideration. We suspect that our method induces representations that are explaining disparate portions of the neural response variance from task-trained models for three reasons. . First,  self-supervised trained models provide neural predictivity that is on par with task trained networks (i.e. supervised recognition networks) [1, 2]. Second, above a certain threshold (in the regime that modern computer vision networks occupy) task performance (top-1 accuracy on ImageNet-1k) is actually negatively correlated with neural predictivity [3]. Finally it is worth noting that after retraining our most predictive model for an increased number of epochs, we now obtain SOTA performance on IT predictivity. However, it would certainly be interesting to more directly answer the question of how “shared” the explained variance is between disparate sets of models, for example by examining the structure of the residuals obtained in the Model-to-Brain regression problem [4]. We could begin to answer some of these questions using new and open source datasets of neural measurements in Macaque IT [5] and will highlight the importance of such investigations in the revised manuscript.
>
> - Writing in Section 3.2: We agree that these experiments could be described more clearly. The idea for a table that describes the sources of variability being compared in each panel is a good one and we will include an Appendix with additional details in the revised manuscript. We will also make the suggested updates to axis labels.
>
> - Thank you for pointing out these typographical issues! We will correct these during the revision process.
>
> - Confusion regarding lines 161-162: Thank you for pointing out the ambiguity in our word choice. There need not be a plural in this sentence, the intended meaning was that we repeated this pairwise comparison (between one invariant and one equivariant network) for each of the equivariant networks we trained (different base objective functions and different values of lambda). We propose to change this sentence to:
>
>      "In particular, we estimate C1 and C2 over identical inputs for an invariant network and an equivariant network trained using the same base objective but a non-zero value of $\lambda$."
>
> - Grey Inset: This was referring to the gray square in Figure 1 and is another typographical error. We apologize for the confusion and will correct this error.
>
>
> [1] Zhuang, Chengxu, et al. "Unsupervised neural network models of the ventral visual stream." Proceedings of the National Academy of Sciences 118.3 (2021): e2014196118.
>
> [2] Conwell, Colin, et al. "What can 1.8 billion regressions tell us about the pressures shaping high-level visual representation in brains and machines?." BioRxiv (2022): 2022-03.
>
> [3] Schrimpf, Martin, et al. "Brain-score: Which artificial neural network for object recognition is most brain-like?." BioRxiv (2018): 407007.
>
> [4] Canatar, Abdulkadir, et al. "A spectral theory of neural prediction and alignment." Advances in Neural Information Processing Systems 36 (2024).
>
> [5] Madan, Spandan, et al. "Benchmarking Out-of-Distribution Generalization Capabilities of DNN-based Encoding Models for the Ventral Visual Cortex." arXiv preprint arXiv:2406.16935 (2024).

---

> > ### Comment · Reviewer_d1rT · 2024-08-13
> > **Reply**
> >
> > Thank you we will improve our score.

---

### Official Review · Reviewer_aTov · 2024-07-16

**Soundness:** 4
**Presentation:** 3
**Contribution:** 4
**Rating:** 8
**Confidence:** 4

**Summary:**

The paper introduces a novel framework, CE-SSL, to address the limitations of traditional self-supervised learning (SSL) objectives, which often result in overly invariant network representations, with a goal to improve the neuronal plausibility of resulting representations. The authors propose a method that incorporates structured variability in response to input transformations, aligning the representations more closely with known features of visual perception and neural computation. The CE-SSL framework converts standard invariant SSL losses into contrastive-equivariant versions, encouraging the preservation of aspects of the input transformation without supervised access to transformation parameters. The method was validated through representational analyses, neural predictivity evaluations using the BrainScore pipeline, and various downstream tasks, demonstrating improved alignment with neural responses and increased structured variability in the representations.

**Strengths:**

Innovative Framework: The proposed CE-SSL framework is a novel approach that effectively addresses the problem of excess invariance in traditional SSL methods by incorporating structured variability, aligning better with biological visual systems.
Biological Plausibility: The method is designed to align more closely with neural responses in the primate visual system, particularly the inferior temporal cortex, which is a significant advancement in the field of neural predictivity.
Comprehensive Validation: The authors thoroughly validate their method through multiple analyses, including representational analyses, neural predictivity evaluations using the BrainScore pipeline, and various downstream tasks.
Practical Implementation: The method does not require supervised access to transformation parameters and involves minimal modifications to the training procedure, making it practical for implementation.
Detailed Analysis: The paper provides a detailed analysis of the tradeoff between invariance and structured variability, offering valuable insights into the representational properties of the learned models.

**Weaknesses:**

Sensitivity to Hyperparameters: The performance of the method is sensitive to the choice of the hyperparameter λ, which controls the balance between invariant and equivariant loss terms, requiring careful tuning.
Limited Generalization: The method showed limited improvement in generalization to out-of-distribution tasks when trained on large datasets like ImageNet-1k, which could limit its applicability in diverse real-world scenarios.
Limited broader impact: The method improves biological plausibility of learned representations, but the broader impact to representation learning in AI/ML is not clear.
Presentation: Some concepts are insufficiently described, for example the projection network architecture is not descibed in sufficient detail. Overall, the organization of the paper is a bit convoluted and more work is needed to streamline the narrative.

**Questions:**

How does the method perform with different backbone architectures other than ResNet-50? Have the authors considered testing with more recent architectures?
How robust is the method to changes in the choice of augmentations? Have the authors tested with different sets of augmentations?
What are the potential applications of the CE-SSL framework in other domains beyond visual perception, such as audio or text processing?
Have the authors considered integrating additional forms of self-supervision, such as clustering or reconstruction, to further enhance the learned representations?

**Limitations:**

The authors have adequately addressed the limitations of their method, including the computational overhead, sensitivity to hyperparameters, and limited out-of-distribution generalization. They have also provided constructive suggestions for future work to address these limitations, such as exploring temporal self-supervised learning and more ecologically relevant data sources. The authors' transparency in discussing these limitations is commendable and aligns with the best practices for responsible machine learning research.

---

> ### Author Rebuttal · Authors · 2024-08-07
>
> Thank you for your careful review of our submission. We respond to the listed weaknesses and questions below:
>
> - Introducing a new hyperparameter to tune: we agree that this is a fundamental weakness of our framework. It would be preferable to balance this tradeoff without using two loss functions and a Lagrange multiplier. We will add the following to the limitations section (Line 524):
>
>     “Our current training setup requires selection of the hyperparameter $\lambda$ to balance between the equivariant and invariant loss functions. Future work could investigate methods to balance the two losses without explicitly training an individual network for each choice of $\lambda$.”
>
> - Performance on OOD Classification tasks: Please see our general response for a more detailed explanation for why equivariance leads to little improvement in OOD classification performance (relative to invariant SSL) when trained on a sufficiently large and diverse set of natural images. While we agree that this in some sense limits the applicability of our method in AI/ML representation learning, we feel that this observation is in and of itself a contribution to the community. A similar effect was observed in some past work on Equivariant SSL (see Table 2 vs Table 3 in [1]), but we feel that it can only benefit the community to draw attention to the interplay between inductive biases (enforced via the loss) and the effect of dataset scaling on learned representations.
>
> - Other backbone architectures: We agree that it is important to confirm that the effects we report are not limited to a specific choice of architecture. In response to this concern, we trained a limited set of models using either ResNet-34 or ResNet-101 backbones in place of the ResNet-50 considered in the text. We found similar effects in terms of the neural predictivity of the equivariant networks relative to their invariant counterparts (see Figure R3 in the rebuttal pdf). We acknowledge that this is not a very radical departure from our initial architecture, but time prevented us from considering more modern backbones such as ResNext and ViT. We will endeavor to include experiments with these backbones in the final version of our paper, and will certainly include the experiments from Figure R3 of the Rebuttal PDF as a new Appendix.
>
> - Robustness to choice of augmentations: This is an interesting question to consider. To begin answering this we conducted experiments involving pretraining models using weaker versions of the same transformations (see general response and Figure R1 of the PDF for more details). Of course these experiments do not entirely answer the question of how robust our method and observations are to the choice of input transformations, though they do suggest a form of generalization (from representing weak-to-strong transformations) is possible. We feel the most important and ecologically relevant set of input transformations are those that mimic the visual experience of animals in the world, and as we note in the discussion we believe this is a very promising direction for future work!
>
> - Other input modalities: Great question - we have been thinking about how this technique might be applied to learning representations of audio signals! In this set-up we envision using a speech-in-background setting: where “positive” pairs would consist of the same speech signal with different choices of background noise. Networks can then be trained to be invariant to background noise, or equivariant to background noise using the method described in this work. While these experiments are beyond the scope of this paper, we plan on following up in this direction with subsequent work. We had not considered applying our method to text, though in view of our analogy between invariance/equivariance and slowness/straightness and the recent observations in [2], this may also be an interesting possibility. We will add the following sentence to the discussion line 296 to capture this idea:
>
>     “Although in this work we focused on the visual domain, similar equivariant and invariant objectives could be investigated for other domains such as audio and langauge representation learning.”
>
> - Other forms of Self-Supervision: In terms of clustering, we believe that existing methods that use clustering based contrastive losses (i.e. SwAV) would likely show similar results to the base objective functions considered in this work. A reconstruction loss would certainly also “fight-against” the invariance term, as it explicitly encourages that information about the input pixels be preserved. It would be interesting to see whether such a scheme also produced more predictive representations of IT, though this may be out of the scope of the current work.
>
> - Presentation: We will also add additional details on projector network architectures to Appendix A.2: Additional pretraining details.
>
> [1] Chavhan, Ruchika, et al. "Amortised Invariance Learning for Contrastive Self-Supervision." The Eleventh International Conference on Learning Representations, 2023.
>
> [2] Hosseini, Eghbal, and Evelina Fedorenko. "Large language models implicitly learn to straighten neural sentence trajectories to construct a predictive representation of natural language." Advances in Neural Information Processing Systems 36 (2024).

---

### Author Rebuttal · Authors · 2024-08-07

We would like to thank the reviewers for their careful consideration of our contributions and thoughtful questions. Several points were raised in multiple reviews and we respond to these below:
1. Lack of Context/marginal improvements on neural predictivity benchmarks: We agree that we provided insufficient context in order to evaluate the effect size of our equivariant training intervention on neural predictivity in IT. To better situate our models performances, we generated histograms that show the performances of all models on the Public Brain-Score leaderboard for each IT dataset considered (Figure R2 in PDF), which we will add to the main text of the paper for the final version. For Barlow Twins, our vanilla model ranked 50th overall and the strongest equivariant model (\lambda = 0.2) ranked 10th overall in terms of mean predicitivity. However, we only trained each ImageNet1000 model in our submission for 100 epochs to reduce computational costs of training many varieties of models (compared to 1000 epochs which the most performant SSL models generally use). We’ve now retrained our best model using twice as many (200) epochs, and the resulting equivariant Barlow Twins model is actually the #1 performing model. In addition to highlighting this via the included figure (Figure R2 in the PDF), we plan to include the following at Line 225 to reflect this:

    _“Additionally, by training the model that best predicts IT data (BarlowTwins, lambda=0.2) for a total of 200 epochs, compared with 100
    for the other models, we achieved 0.5330 mean fraction of explained variance, which makes this the top IT brain prediction model.”_

2. Dependence on augmentation distribution: We chose to use the standard set of augmentations employed in modern self-supervised learning schemes as these are known to yield models that are performant on downstream tasks and predictive of neural responses, but we agree that considering the sensitivity of our method to augmentation choices is worthwhile. To investigate this with our framework, we trained a new set of models using uniformly weaker augmentations and the Barlow Twins objective (specifically, we doubled the minimum size of random crops, halved the maximum values of color jitter modulations, and halved the maximum std of gaussian blurring), and repeated the parameter decoding experiments from Figure 2E. The results are summarized in Figure R1 of the attached PDF. We find evidence that models trained on weak augmentations do still exhibit structured variability that generalizes to stronger input transformations. We will include this result in the Appendix, and the following at line 217:

    _"We further analyzed a set of equivariant models trained with weaker augmentation parameters (Figure R1) In these networks, we once again observe that equivariant training increased the amount of linearly accessible augmentation information compared to invariant training. This is the case not only for the augmentation parameters they were trained on (left panel) but also for parameters beyond the range of the training distribution (right panel). This suggests that the models represent some equivariance beyond the training distribution, and future work could further investigate the interaction between equivariant training and improved generalization to unseen types of augmentations."_

3. Performance on downstream classification tasks: We believe that multiple factors could contribute to our observation that encouraging equivariance during pretraining has little effect on OOD downstream performance for invariant classification tasks, and can actually harm performance on the in-distribution setting.  One important factor is the design of the augmentations used in this work (and widely across invariance based self-supervised learning methods). This particular set of transformations was refined by the community in order to increase the task alignment between the self-supervised learning and supervised object classification (i.e., the augmentation distribution was tweaked in order to maximize the downstream classification performance on ImageNet). In light of this we actually expected the in-distribution performance on classification to be reduced, and find it notable that in many settings the performance penalty is quite small (< 3%), while still inducing a large amount of new structure in the representation.

    However, for out-of-distribution classification tasks where the task-alignment is worse, the equivariance task could mitigate this mis-match. A concrete example is the Flowers-102 dataset, where the color of petals is a much stronger predictor of class than color is in, say, the ImageNet-1k dataset (so the color insensitivity induced by the standard augmentations could be detrimental). For this dataset we do see marginal improvements, but note that the improvements are much more pronounced when pretraining on smaller datasets (ImageNet-100). There are 2 possible explanations for this: (1) for ImageNet-1k pretraining the performance of the networks is already quite high, and the task is saturated, or (2) there is a more fundamental reason that the improvements in transfer learning induced by equivariance decrease as the size and diversity of the pretraining dataset grows. Our suspicion is that factor (1) is actually the dominant cause, but we certainly agree that disambiguating between these possibilities would be worthwhile. We plan to describe this issue in the Discussion section of the revised manuscript to better highlight these observations.

4. Overall presentation and typographical errors: Thank you for highlighting parts of the paper that were confusing or incorrect. The typos will be corrected for the final version of the manuscript, and we'll make another pass through the text to improve readability, especially for Section 2 describing the methods and section 3.2 describing the representational distance experiments.

---

### Decision · Program_Chairs · 2024-09-25

**Decision:**

Accept (poster)

**Comment:**

This paper describes a novel SSL approach incorporating constraints for learning less invariant and more equivariant representations.

The reviewers unanimously found the proposal novel and worthy. One main limitation of the work is the seemingly limited improvements in OOD classification compared to current methods. Even in the case of brain predictability, one could argue that the improvements are marginal. Nonetheless, the work appears to be carried well, and there is unanimous agreement that the approach is moving the field in the right direction.

Hence, the AC recommends the paper to be accepted.